# Quality of Life among Caregivers of Patients Diagnosed with Major Chronic Disease during COVID-19 in Saudi Arabia

**DOI:** 10.3390/healthcare10030523

**Published:** 2022-03-13

**Authors:** Mohammed Aljuaid, Namrah Ilyas, Eman Altuwaijri, Haddel Albedawi, Ohoud Alanazi, Duaa Shahid, Wadi Alonazi

**Affiliations:** 1Department of Health Administration, College of Business Administration, King Saud University, Riyadh 11587, Saudi Arabia; maljuaid@ksu.edu.sa (M.A.); ealtuwaijri@ksu.edu.sa (E.A.); 2Centre for Clinical Psychology, University of the Punjab, Lahore 66000, Pakistan; namrah.sindhu@hotmail.com; 3Department of Administrative and Social Sciences, College of Business Applied Studies and Community Service, King Saud University, Riyadh 11587, Saudi Arabia; 4Department of Community Health, College of Applied Medical Sciences, King Saud University, Riyadh 11587, Saudi Arabia; hadeelbed@gmail.com; 5Department of Public Administration, College of Business Administration, King Saud University, Riyadh 11587, Saudi Arabia; oa-alenazi@ut.edu.sa; 6Hult International Business School, Cambridge, MA 02141, USA; duaa.shahid@hult.edu

**Keywords:** chronic disease, heart disease, cancer patients, diabetes patients, quality of life, caregivers

## Abstract

Quality of life (QoL) is considered one of the measures of health outcomes. Limited research studies have assessed family caregivers’ QoL, especially among patients diagnosed with chronic disease. This study measures the QoL of caregivers who guardian patients diagnosed with cardiovascular disease, diabetes, cancer, and/or other diseases during the COVID-19 pandemic. Participants were primary caregivers who were supporting, in the last six months, individuals diagnosed with one of the previously mentioned chronic diseases. This included caregivers of patients admitted to a tertiary hospital from January 2021 to July of the same year (*n* = 1081); all participants completed the World Health Organization Quality of Life Assessment tool (WHOQOL-BREF) questionnaire. Caregivers of patients with cancer reported the highest mean level of QoL, followed by diabetes, cardiovascular diseases, then other different diseases (M = 3.80; M = 3.38; M = 3.37; and M = 2.51, respectively). A chi-square test of independence was performed to examine the relationship between the QoL of the four groups and their behaviors (i.e., caregivers’ psychological onuses and physical actions/reactions). The relation between these variables was significant, X^2^ (3, *n* = 1081) = 8.9, *p* = 0.001. The Kruskal–Wallis test indicated significant differences among the four groups (*p* ≤ 0.001). While the overall results of the QoL level of participants were low, a major recommendation of this study was to incorporate a QoL assessment to caregivers of chronically ill patients. Regular psychological and physical health check-ups of caregivers should be mandated in the healthcare system. Research studies should consider investigating and identifying the factors affecting health outcomes and positive developments which have a great impact on the wellbeing of both caregivers and patients on personal, organizational, and national levels.

## 1. Introduction

This is a universal truth that the high life expectancy rate leads to an increase in the elderly population, therefore upsurging the demand for caregivers all around the world at large and specifically in Saudi Arabia. Comparatively, Saudi society inter alia Gulf Cooperation Council (GCC) nations is quite conservative and has a high inclination to follow their traditions (rituals and social norms). These societal values encourage primary caregivers (registered family members and close relatives) to provide care to their long-lasting unwell ones [1,2]. Subsequently, the Quality of Life (QoL) of caregivers is impacted in respect of the social, physical and psychological aspects (i.e., obligation, daily routine, stress, distress, lack of sleep), and ultimately they suffer from substantial health issues [3]. However, most studies were conducted about primary caregivers of the patient(s) diagnosed with chronic disease who go through different levels of health risks, including QoL but in a Western context. An extensive review of the literature sheds little light on the healthiness of vigilances from the perspective of GCC nations, specifically their QoL in a Saudi context [4,5].

While many research studies have focused more on the health evaluation of organizational resources and activities [1,2], the QoL of caregivers and experiences are normally limited in the literature. Indeed, many studies have attempted to eliminate the interference and preferences of caregivers while measuring the healthcare system, particularly in chronic diseases such as cancer, diabetes, and cardiovascular diseases [6,7].

According to the National Center for Chronic Disease Prevention and Health Promotion (NCCDPHP), chronic diseases are defined broadly as conditions that last one year or more and require ongoing medical attention and/or limited daily activities. Chronic diseases such as cardiovascular diseases, cancer, and diabetes mellitus (DM) are the leading causes of death and disabilities worldwide. Their health care cost in the USA is estimated to be USD 3.8 trillion annually. Chronic diseases are not only affecting patients themselves independently, but they also have an impact on their families and individuals around them. Generally, caregivers are sacrificing their QoL and facing enormous burdens in terms of physical and non-physical manners. It is quite obvious that caring about a patient with a chronic cardiovascular disease, for instance, is both demanding and rewarding. In the same vein, it may have a great impact on the life of the caregiver in many aspects, whether it is physical, psychological or/and emotional.

It is a challenge to provide care to a patient with cardiovascular problems as they might undergo life-threatening incidents unexpectedly, as it is not an easy task to perform as other regular jobs. During the first year, the overall wellbeing of a caregiver improves and reaches up to normal; on the other hand, after twelve months of care, caregivers with emotional or cognitive problems are at risk of developing a higher care burden [8,9]. Normally, caregivers providing care to a cardiovascular patient with depressive symptoms have higher levels of caregiver burdens and low QoL compared to those providing care to patients without depressive symptoms [10]. Patients with cardiovascular diseases may influence the caregiver experience by reducing their QoL with a high incidence of depression. Inclusively, caregivers may experience depression symptoms due to lower patient QoL, disease burden, and their own caregiver burden [11]. Family and caregivers with such complex cardiovascular diseases are emotionally and psychologically affected; this influences their care, behaviors, and attitudes. Specifically speaking, the emotional challenges that caregivers face and the practicalities of undertaking the role of a caregiver has an impact on both the caregiver and the QoL of the patient [12]. Most commonly, care provided by the caregivers can be experienced as being stressful. In addition, caregivers provide financial, emotional, and physical support for always being on-call even when having obligations [13,14]. Furthermore, compared with non-caregivers, caregivers are more vulnerable to more chronic conditions and health problems, need time off from work, and are normally associated with a lack of a health insurance policy, according to the previous study.

Several authors investigated the QoL, the psychological stress, the unmet needs, and satisfaction with care of family caregivers of advanced cancer patients receiving specialist inpatient palliative care. Caregivers were systematically recruited within 72 h of the patient’s arrival. Family caregivers of chronically ill patients at home were reported to have a number of physical health difficulties, including fatigue and sleep disruptions [15,16]. Contrary, cancer caregivers in Saudi Arabia have reported favorable results of their QoL. They showed better functioning on QoL domains of emotional wellbeing, role functioning/emotional, pain, social functioning, physical functioning, general health, and role/physical functioning [17,18]. The energy fatigue was the lowest score among the QoL domains indicating that caregivers may be at risk of poor QoL when time and disease advance [19]. As for physiological challenges, some studies disclosed that many caregivers did not have time to seek medical help if having an injury, or they may neglect their own need for healthcare, including treatment of some health conditions such as diabetes or high blood pressure. Most people reported a low level of QoL, especially when they take care of cardiovascular and diabetes patients [18,19,20,21].

An extensive review of the literature indicated a vast number of studies from the perspective of a Western context of the QoL balance among caregivers of chronic disease patients. On the other hand, little research is associated with anecdotal accounts in GCC and regional countries to examine the relationships between the QoL of caregivers and their behaviors. Such behavior may have been proposed from time to time as a critical factor to coping with a person affected with chronic illness. However, the literature in the context of this study is less endowed with descriptions, explanations, explorations, and predictions [22,23]. This empirical study intended to describe and explain the singularity. i.e., why the QoL of caregivers’ psychological obligations and physical actions/reactions are not investigated especially in the course of a pandemic. Therefore, there is a dearth of studies in emerging economies such as GCC and specifically in a Saudi context to understand the caregivers’ behaviors during the period of a pandemic to maintain their QoL while taking care of chronically ill patients.

### Assessing Quality of Life in Saudi Arabia

As mentioned earlier, the key aim of this study is to investigate the QoL of primary caregivers who support their dependents that are suffering from a chronic disease and are under hospital treatment. In the same vein, QoL refers to the appraisal of and satisfaction with caregivers’ current level of functioning compared with what they normally exercise. This concept encompasses a broad range of physical and psychological characteristics and limitations that describe the ability of caregivers to function and to derive satisfaction from being a close helper. The World Health Organization (WHO) measures had focused on the perceptions of individuals of their position in life in the context of culture and systems in which they live, in addition to the relation to their goals, expectations, standards, and concerns. A number of empirical and non-empirical studies that incorporated WHO measures are most appropriate as they describe the core components of health and wellbeing that can be practically applied in any type of health context [24,25,26,27].

In order to briefly explore QoL, some concepts need to be discussed about the need for physical, social, psychological, environmental, and overall health satisfaction assessment. Again, caregivers are considered to be one of the stressful but greatest humanitarian missions in the community. Providing care to everyone in need would result in struggling with high stress causing “Caregiver Burden”, which affects their physical, mental, and wellbeing [28]. A study in the East Region of Saudi Arabia about this condition on the elderly showed that 65% are exposed to caregiver burden, 15% of them were considered as a severe burden [29]. Alqahtani and colleagues conducted another empirical study to find out the causes or effects of the prevalence of depression among caregivers of Alzheimer’s disease in Saudi Arabia. The results of this research indicated the caregivers’ levels in stress, depression, and anxiety are significantly high [30]. However, the provision of social support systems for caregivers may help improve their QoL and help with the challenges they are encountering.

Hence, the high prevalence of burden on caregivers should be noticed and tackled by the provision of education and effective financial and mental support. This paper intends to determine the QoL of caregivers of patients diagnosed with cardiovascular diseases, diabetes, cancer, and other diseases during COVID-19.

This study attempts to address the caregivers’ physical, emotional, psychological, and social wellbeing to maintain their QoL in the course of the COVID-19 pandemic in Saudi Arabia. The primary aim of this paper is to draw the attention of healthcare researchers, professionals, policymakers, and practitioners by presenting the insights that have gleaned from this empirical study.

## 2. Materials and Methods

The design of this study is descriptive, which is followed by the STROBE guidelines to unveil whether chronic disease patients affect the QoL of caregivers during COVID-19. This design is considered most appropriate due to the nature of this empirical research. It also structures the major parts of the study, i.e., the sample, measures, and methods for data collection and analysis to answer the central question of interest.

### 2.1. Tool

A survey measurement tool was used and administered in the form of a questionnaire, which was self-administrative in order to gather data from the respondents. The reliability and validity include psychometric properties of each element rigorously rechecked at the finalization stage of the measurement instrument (questionnaire) by the panel of experts. The World Health Organization Quality of Life Assessment tool WHOQOL-BREF was used in this study with minor modifications as demanded by the context. This tool is highly recommended for use in large epidemiologic surveys and clinical trials where time is restricted; detailed information is unnecessary with limited resources [6]. The researchers grouped the measurement instrument into four QoL domains (physical health, psychological health, social relationships, environment, and two supplementary items, i.e., overall QoL and general health, to accomplish the set objectives).

After the approval of the Institutional Review Board (IRB), a number of head nurses were approached for consent to assist researchers in accessing respondents who hold the answer to the key question of this study. The inter alia five individuals granted their consent to act on behalf of researchers as a resource person to collect data via an online survey. The research team trained these nurses on the contents of the research instrument in-person to gather data in alliance with the appropriate ethical protocols. The data collection tool contained direct questions and the average time to complete a questionnaire was between 25 and 30 min. The final sample population was 2000, of whom 1081 responded fully to the survey questions.

### 2.2. Settings

Five major hospitals in the metropolitan city of Riyadh, Saudi Arabia, offer tertiary services to a population of five million people.

### 2.3. Subjects

Subjects of the study were caregivers of dependents who were diagnosed and treated for cardiovascular disease, diabetes, cancer, and/or other chronic diseases (kidney disease, Alzheimer’s disease, and lower respiratory infections). Participants were 2000 individuals who were invited to participate voluntarily in the study and were skillful to access and answer an online survey. The set scientific criteria were conducted for the selection of the study participants. Participants were randomly selected; this is to give each member of the population the likelihood to be selected for the research sample.

### 2.4. Data Collection

The data collected from the public sector hospitals started in January 2021 and ended in July 2021, and some selected head nurses identified the caregivers of patients who were admitted to the hospital and were suffering from chronic diseases. Data were collected via Google Forms through a secured and official link. The head nurse was the only individual to contact caregivers who spent at least six months with their dependent diagnosed with a chronic illness.

### 2.5. Statistical Analysis

Statistical Package for Social Sciences (SPSS) software was used to calculate descriptive analysis (mean, standard deviation, and percentage). A matrix of correlation, Kruskal–Wallis and other advanced statistical tests were applied to predict the inferential analyses.

## 3. Results

The demographic characteristics of participants are described in Table 1. Out of 2000 selected participants, only 1081 participated in this cross-sectional study. The total response rate was 54.05%, of which 59.7% were males, and 40.3% were females, 68.8% of participants stated that their marital status was single. Thirty-four percent of the primary caregivers were between the age of 21 and 30 years, which reflects this age group of the young generation towards their obligation, societal values, and norms. It was also observed that 59.9% of participants reached an educational level of high school or below. Moreover, 50.8% of participants’ monthly income was less than 5000 SAR per month, which is below the average income compared to the income of the middle class; 31% of the participants and 8.3% of participants of the upper-middle class or above. In terms of primary caregivers, 39.4% of participants were mothers who were involved in this process of care. Finally, 50.2% of chronic patients that were being taken care of are suffering from a cardiovascular disease inter alia.

The QoL of caregivers assessed by WHOQOL-BREF is shown in Table 2. The data show that half of the participants reported their QoL to be good or very good (51.4%), while 31.8% were neutral. Likewise, 49.2% reported that they are neither satisfied nor dissatisfied with their overall health. As for the domains of WHOQOL-BREF, the mean score for the social relationships was the highest (61.56 ± 15.89), while the lowest mean score was for physical health (56.07 ± 7.54).

The behavioral characteristics of participants are shown in Table 3. Overall, one-fourth of the respondents were smokers, 23.2% were exercising at least three times a week, and 46% reported following a balanced diet, while 59.2% strictly followed the pharmaceutical instructions. Furthermore, more than half of participants reported having a good knowledge of current health issues, strong social networks, and encouraged their families to become vaccinated (63%, 62.4%, and 68.5%, respectively). The association between behavioral characteristics of caregivers and the type of disease was measured using Chi-square. There was a significant difference in all behavioral variables across different types of diseases.

The correlations between overall QoL, health satisfaction, and WHOQOL-BREF domains are shown in Table 4. The data show a moderate positive correlation between QoL and health satisfaction (r = 0.469, *p* < 0.001). In the same manner, the domain of social relationships positively correlated with the domains of health satisfaction and psychological health (r = 0.497, *p* < 0.001, r = 0.494, *p* < 0.001, respectively) whilst the domain that had the strongest correlation was between environment and physical health (r = 0.529, *p* < 0.001).

The difference in the WHOQOL-BREF scores across different chronic diseases is shown in Table 5. The Kruskal–Wallis test showed a significant difference in most domains of WHOQOL-BREF. A pairwise comparison shows that caregivers who cared for cancer patients had significantly higher scores in their overall QoL (in both the psychological and social domains) compared to those of patients with cardiovascular diseases.

In the same way, caregivers of cancer patients also had significantly higher scores in overall QoL (in both the health satisfaction and social domain) when compared to caregivers of diabetes patients. On the other hand, caregivers of individuals with other chronic diseases had significantly lower overall QoL (in both health satisfaction and the social domain) but a higher score in the physical domain compared to caregivers of cardiovascular diseases, diabetes, and cancer patients.

The Kruskal–Wallis test was used to determine the difference in WHOQOL-BREF scores across the different chronic diseases. The associations between WHOQOL-BREF domains and the demographic characteristics of participants were assessed using Mann–Whitney U test (Table 6). Females had significantly higher scores in QoL, health satisfaction, and social relationships, whereas males had significantly higher scores in physical health and environment (*p* < 0.001, for all). Similarly, individuals at the age of 40 years or younger had significantly higher scores in their QoL, health satisfaction, and social relationships, while older individuals (>40 years) had significantly higher scores in physical and environmental domains. Based on education, the QoL, health satisfaction, psychological and social domains were significantly higher in participants with at least an undergraduate degree compared to participants with a lower level of education (*p* < 0.001, *p* < 0.042, *p* < 0.018, *p* < 0.014, respectively). Marital status has also shown a significant difference in physical, social, and environment domains with lower scores for married individuals (*p* < 0.001, *p* < 0.010, *p* < 0.001, respectively). Similarly, participants with higher monthly income had significantly higher health satisfaction, physical, psychological, social and environment domains (*p* < 0.007, *p* < 0.001, *p* < 0.001, *p* < 0.002, *p* < 0.001, respectively). On the other hand, smokers showed a significantly higher scores in QoL, health satisfaction, and social and environment domains (*p* < 0.001, *p* < 0.038, *p* < 0.001, *p* < 0.001, respectively).

## 4. Discussion

The present study aimed to assess the level of QoL and its domains among caregivers of patients diagnosed with chronic diseases. Caregivers who take care of cancer patients had the highest level of QoL, followed by diabetes, cardiovascular diseases, then other diseases. The overall results of this study are still consistent with some previous research studies [9,16,17,18].

With the expansion of chronically ill patients in the population, more ill individuals depend on their families to take care of them. Despite the burdensome experience of caregiving, the QoL of the study participants showed was good overall. However, caregiving is a life-altering journey for many caregivers. Anderson and White identified three major rewarding themes of the caregiving experience [31]. This includes: (1) the gratitude for being able to help those in need, (2) their sense of accomplishment, and (3) realization of the significance of good health. These positive experiences faced by caregivers positively influence their wellbeing and enhance their QoL.

Thus, caregiving is a transforming experience for individuals. It empowers them to overcome different challenges, instill resilience as well as develop stress management and coping strategies [32]. Cancer is a disease that impacts the whole family during the illness trajectory. Caregivers go through various emotions and challenging situations [33]. Despite the stressors involved in the caregiving process, Kim, Schulz, and Charles identified six benefits of the experience manifested by caregivers of cancer patients [16]. This includes acceptance of events, empathy towards others, appreciation of the newly developed relationship with loved ones and family, positive self-view, and reprioritization of goals and aims in life. These are bigger and wholesome goals that lead to a wider perspective of appreciation and, in the long run, will have a positive impact on the QoL of caregivers [34].

Another group that appeared to have good QoL in the current sample is caregivers of cardiovascular patients. Cardiovascular diseases are yet another debilitating chronic disease that requires monitoring and care during illness [35]. Caregiving to a cardiovascular patient leads to the burden, stress, and poor QoL across the physical, emotional and social domains [36]. That said, certain factors can contribute to the improvement and/or maintenance in the QoL of caregivers of cardiovascular patients. Lum et al. identified that good relationships are associated normally with caregivers of patients with cardiovascular patients [37]. Furthermore, social support, stage of illness, and the mental health of patients play a significant role in the QoL of caregivers [38].

The caregivers of diabetics had the third-best score on their QoL, even though diabetes is a lifelong chronic disease requiring self-management and continuous care from members of family and friends [39]. However, Awadalla et al. identified various factors that impact the QoL of caregivers of diabetics [40,41]. The type of diabetes is essential in determining the QoL for both the caregiver and the diabetic person. It might be suggested, based on the evidence of this study, that various protective factors have positive impacts on the QoL of caregivers of diabetic people. The findings of this study also indicated the impact of the demographic variables on the QoL of the participants were inconsistent with previous studies from the perspective of the Saudi context.

Results from our study contradict what is found in the literature of varying estimates across different countries where the female gender predominates caregiving, as 57–81% of caregivers of chronic patients are females [42]. However, there was not a significant difference between both genders, and we can attribute the minor variance to factors including the role of males to fulfill the basic needs of the family and perform the outside minor and major tasks, including hospital visits. Thus, the male gender could be a little over-represented in the sample. Moreover, there were no differences in gender distribution; this could be attributed to the fact that males in cultural norms in Saudi. Arabs are more involved in the family’s wellbeing.

Furthermore, most participants had a balanced diet, strong social ties, had the latest knowledge of health issues, and encouraged their families to become vaccinated. This suggests that caregivers are well-aware of how to take care of their health and take active steps towards instruction by pharmaceutical companies, following a healthy diet, and encouraging family members to take care of their health. Indeed, a demanding caregiving role leads to depression, psychological distress, poor self-care, and poorer self-reported health [43,44]. Certain health-related behavior factors can determine the adverse outcomes of caregivers depending on their perception of stress and the burden of care.

Therefore, it is reasonable to assume that caregivers do not feel burdened despite the chronicity of illness of their patients. This factor can also be confirmed by the overall QoL of the caregivers and health satisfaction. Almost half of the caregivers reported that their QoL was good or very good. Additionally, nearly half of the participants were neither satisfied nor dissatisfied with their overall health. Thus, participants were neutral regarding their physical health status. Moreover, the mean score on all domains of QoL was above 50, suggesting a good overall QoL in all domains of the scale [45].

The caregivers are at high risk of deteriorating their QoL as they spend more time and energy taking care of chronically ill patients. However, some caregivers indicate fulfillment in the role of caregiving even when the burden is high; this is due to family relationships, guilt, or duty [32]. Based on large population-based studies, Sherwood and Schulz suggested that almost one-third of caregivers experienced neither strain nor any negative health effects because of the caregiver role. Even with the fact of increased caregiving demands, caregivers often found positive aspects of the experience. It is reported that caregiving gives them a purpose and meaning in life, makes them feel good about themselves, allows them to adapt to situations, and strengthens their relations with others [46,47]. Moreover, it is evident from research studies that supportive social relationships promote happier and healthier lives compared to socially isolated individuals [48]. Research findings also suggest that supporting or helping others is as advantageous to health as receiving help. Brown et al. [49] found that individuals who support their family, friends, or neighbors, and people who give emotional support to their significant others or spouses had better health, lower mortality rate, and were satisfied with their quality of life. The demographic correlates of QoL and its domains, including gender, age, education, marital status, occupation, monthly income, and smoking, were assessed. The results showed interesting findings. Some of these findings may contradict the existing literature, and some may add new information. It is because QoL is assessed in different studies by employing different QoL-related theoretical frameworks and definitions. Thus, it is not easy to precisely support our study results with previous research studies.

Results suggested that females had better overall QoL and health satisfaction; they also had more fulfilling social relationships. On the contrary, males had better physical health and scored higher in the environment domain. There can be various possible reasons for these differences among both genders. The literature explicitly suggests that females are vulnerable to low quality of life and poor health because of the caregiving burden [50,51]. However, Broj et al. suggested that male and female caregivers face this caregiving experience differently [52]. They conducted a longitudinal study to assess the trajectory of QoL in both genders within one year. The results suggested that although females experienced a higher burden, the QoL of male caregivers declined significantly compared to females after one year. It can be suggested here that females are good at adapting to the role of caregiving while males find it difficult.

Moreover, while observing the variance in the QoL domain scores across gender, it can be proposed that the better social relationships of females help them to cope with the stresses and strains of caregiving. Males’ scorings were high on physical and environmental domains, yet they had lower QoL and health satisfaction; this indicates the fact that caregiving is a burdensome task for them. Lopez and Kohli found that male caregivers experienced emotional stress, financial strain, and had minimal training in taking care of the personal needs of the patient [53]. These factors greatly contribute towards the lower wellbeing of male caregivers as compared to females.

Participants younger than 40 years showed better QoL and health satisfaction. Furthermore, this age group scored better in the psychological and social domains. On the contrary, participants older than 40 years scored better on the physical and environmental domains. The older the age group is, the lower QoL and health satisfaction are; this can be linked to the aging process of individuals. Basheer also found that the QoL of caregivers is inversely related to their age. This can be attributed to old age-related morbidities and financial dependence [54]. Moreover, along with caregivers’ health issues, they have to support an ill person emotionally and be responsible for their daily life activities, which can be burdensome. Furthermore, Hadryś et al. suggested that older caregivers experience a higher caregiving burden, thus compromising their QoL [55]. Therefore, the aging itself can make the caregiving process stressful, compromising the QoL and health of the senior age group. Furthermore, in Saudi Arabia, people live in extended families where the eldest male member of the family is considered the head of the family with ample responsibilities. The other younger family members rely on him for advice and help in daily matters. Among all, taking care of an ill person in a family can be heart-wrenching [56]. Furthermore, family ties in Saudi Arabia are very strong, and family members value them greatly. The sentiments and emotional associations increase with older age, making a person vulnerable to lower QoL and psychological problems.

On the other hand, the younger group reported better QoL and health satisfaction mainly because they have better social relationships, are energetic, and have better psychological health. Moreover, they may feel obligated to take care of the chronically ill patient if they share blood relations, i.e., the patient is a spouse, parent, parent-in-law, or any other elderly family member.

Our findings also showed that participants with a bachelor’s degree or above had better QoL, health satisfaction, psychological health, and social relationships. While Basheer et al. found a similar trend in their study where participants with higher education scored higher on all four domains of QoL, Tasi and colleagues found similar results. Higher education led to a better understanding of the illness and its consequences [54,57]. They can navigate efficiently through the health care system and facilities to make informed decisions regarding medical care and rehabilitation. Furthermore, they have better coping strategies and can adapt better to their role of caregiving [58,59]. Thus, lower levels of education can increase the stress of caregivers resulting in lower QoL in all domains. Single or divorced participants scored better on physical health, social relationship, and environment domains than the married participants, and the difference is significant. The possible explanation of this result is that the caregiving of a terminally ill patient adds a burden to individuals who are already committed to family responsibilities, as the married person has to take care of the family and the ill person [60]. On the contrary, when the caregiver is single, they can direct their energies and undivided attention towards the patient. A systematic review concluded that research studies had yielded contradictory results concerning the impact of the marital status of a caregiver on their QoL. This can be attributed to the cultural differences and family and individual differences in relationship maintenance and development [61]. The employment of caregivers had no significant impact on the overall QoL, health satisfaction, and QoL domains. However, the results suggested that monthly income or economic status are positively related to QoL. The collectivism of the Saudi culture can explain these contradictory findings. Saudi families are closely knitted, and the bread earner is responsible for the entire expenses of the household. Caregivers who responded to the study are supported by the head of the family [60].

Participants with basic financial support reported significantly better health satisfaction and scored higher on all QoL domains. These findings are in accordance with the previous research studies and literature. Again, Jeong et al. reported a positive relationship between the QoL of caregivers and their socioeconomic status, suggesting that financially sound caregivers have a better quality of life [58]. The financial status helps the caregiver to select the best medical care and rehabilitation plans for their patients. They can also have access to various treatment options. Moreover, the caregiving burden does not strain their finances, and they have a sense of accomplishment rather than burdened someone. On the contrary, caregivers with limited financial resources find it difficult to care for the medical expenses of the patients while at the same time supporting their families. This can compromise their QoL and burden caregiving experience [54,59,62]. Another interesting finding of this study is that smokers had better overall QoL and health satisfaction and scored higher on the environment and social domains. The study suggested that smoking is negatively related to QoL. However, in our current sample, the results are the opposite. Smokers only represent one-fourth of the total sample, implying that there can be possible bias. An equal sample of smokers and non-smokers can give us a better understanding of these results.

In summary, the overall QoL of caregivers for chronically ill patients was good in our study sample. The demographic variables including gender, age, education, economic status, marital status, and smoking were significant correlations of QoL and its domains. One of the key objectives of this study, which is based on a large-scale population in Saudi Arabia, was assessing the QoL of caregivers taking care of patients with various chronic medical conditions.

Instead of employing the single construct QoL questionnaire, this study used WHOQOL- BREF that assesses the various domains of QoL. The results, thus, displayed rich information that can be used in public health settings. The limitations of this research study are that the duration of illness, caregiving period, and severity of illness were not specified. Future studies can focus on various other chronic diseases from the perspective of the QoL of caregivers, as mentioned earlier. This present study has implications in public health and medical settings. Caregiving groups with poor QoL, i.e., males, low-income, lower education, and married, should receive more attention from the healthcare system. Support groups should be designed for caregivers that are easily accessible to enhance their QoL, which eventually promotes the health of the patients.

## 5. Conclusions

In this study, we applied a cross-sectional approach to gain insight into the QoL of caregivers of dependents with chronic diseases. Saudi healthcare agencies aim to minimize risks and ensure continuity of healthcare services in the context of this study. Caregiving is a burdensome responsibility, and with the increase in chronic medical conditions in the population, many patients rely on family caregivers. This study assessed the QoL and its demographic correlates in caregivers of chronically ill patients. The results of the study add valuable information to the existing literature. Participants’ mean score on all domains of QoL suggested that they have good QoL because of the density of the social network, specifically in the local society and at large in the country. Moreover, the sample had a higher percentage of male caregivers, which contradicts the literature but can be explained in terms of the male-dominated culture in Saudi Arabia.

## Figures and Tables

**Table 1 healthcare-10-00523-t001:** Demographic characteristics of participants.

Variable	N	%
Age		
20 years or younger	70	6.5
21–30	367	34
31–40	193	17.9
41–50	125	11.6
51–60	151	14
61 years and above	175	16.2
Gender		
Male	645	59.7
Female	436	40.3
Marital status		
Married	289	26.7
Single	744	68.8
Divorced	48	4.4
Education		
No education	337	31.2
High school or less	647	59.9
Bachelors	90	8.3
Masters	7	0.6
Occupation		
Government sector	519	48
Private sector	223	20.6
Businessperson	263	24.3
Unemployed	76	7
Monthly income		
No income	107	9.9
Less than SAR 5000	549	50.8
Between SAR 5000 and 10,000	335	31
More than SAR 10,000	90	8.3
Taking care of Spouse	248	22.9
Father	239	22.1
Mother	426	39.4
Relative	23	2.1
Friend	136	12.6
Other	9	0.8
Suffering from		
Heart diseases	543	50.2
Diabetes	177	16.4
Cancer	290	26.8
Others	71	6.6

**Table 2 healthcare-10-00523-t002:** The quality of life of caregivers assessed by WHOQOL-BREF.

Variable	N = 1081 (%)
Overall quality of life	
Very poor	134 (12.4)
Poor	47 (4.3)
Neither poor nor good	344 (31.8)
Good	332 (30.7)
Very Good	224 (20.7)
Overall satisfaction with health	
Very dissatisfied	31 (2.9)
Dissatisfied	67 (6.2)
Neither satisfied nor dissatisfied	532 (49.2)
Satisfied	266 (24.6)
Very satisfied	185 (17.1)
Overall QoL	
Health Satisfaction	3.47 (0.94)
Physical	56.07 (7.54)
Psychological	57.32 (13.0)
Social	61.56 (15.89)
Environment	58.75 (12.10)

**Table 3 healthcare-10-00523-t003:** Behavioral characteristics of caregivers.

Variable	Overall	Heart Diseases	Diabetes	Cancer	Others	*p*-Value
Smoking						
Yes	270 (25.0)	140 (51.9)	29 (10.7)	96 (35.6)	5 (1.9)	< 0.001
No	811 (75.0)	403 (49.7)	148 (18.2)	194 (23.9)	66 (8.1)	
Exercise						
Yes	349 (23.2)	175 (50.1)	55 (15.8)	111 (31.8)	8 (2.3)	<0.001
No	732 (67.7)	368 (50.3)	122 (16.7)	179 (24.5)	63 (8.6)	
Health Education						
Yes	681 (63.0)	341 (50.1)	93 (13.7)	195 (28.6)	52 (7.6)	0.003
No	400 (37.0)	202 (50.5)	84 (21.0)	95 (23.8)	19 (4.8)	
Diet						
Yes	497 (46.0)	263 (52.9)	64 (12.9)	164 (33.0)	6 (1.2)	<0.001
No	584 (54.0)	280 (47.9)	113 (19.3)	126 (21.6)	65 (11.1)	
Medication						
Yes	640 (59.2)	330 (51.6)	85 (13.3)	208 (32.5)	17 (2.7)	<0.001
No	441 (40.8)	213 (48.3)	92 (20.9)	82 (18.6)	54 (12.2)	
Vaccination						
Yes	741 (68.5)	388 (52.4)	105 (14.2)	216 (29.1)	32 (4.3)	<0.001
No	340 (31.5)	155 (45.6)	72 (21.2)	74 (21.8)	39 (11.5)	
Strong Social networking						
Yes	675 (62.4)	347 (51.4)	108 (16.0	201 (29.8)	19 (2.8)	<0.001
No	406 (37.6)	196 (48.3)	69 (17.0)	89 (21.9)	52 (12.8)	

Pearson Chi-square test was used to determine the difference in behavioral characteristics across different chronic diseases.

**Table 4 healthcare-10-00523-t004:** The correlations between WHOQOL-BREF domains.

	Overall QoL	Health Satisfaction	Physical	Psychological	Social
Health Satisfaction					
Correlation coefficient	0.469				
*p*-value	<0.001				
Physical					
Correlation coefficient	0.080	0.200			
*p*-value	0.008	<0.001			
Psychological					
Correlation coefficient	0.346	0.408	0.233		
*p*-value	<0.001	<0.001	<0.001		
Social					
Correlation coefficient	0.443	0.497	0.244	0.494	
*p*-value	<0.001	<0.001	<0.001	<0.001	
Environment					
Correlation coefficient	0.042	0.311	0.529	0.370	0.330
*p*-value	0.172	<0.001	<0.001	<0.001	<0.001

**Table 5 healthcare-10-00523-t005:** The difference in the WHOQOL-BREF scores across different chronic diseases as assessed by Kruskal–Wallis test followed by pairwise comparison.

Variable	Heart Diseases (1)		Diabetes (2)	Cancer (3)	Others (4)	*p*-Value
Overall QoL	3.37 (1.21)		3.38 (1.17)	3.80 (1.15)	2.51 (1.19)	<0.001
Health Satisfaction	3.49 (0.90)		3.33 (1.11)	3.64 (0.93)	2.94 (0.58)	<0.001
Physical	55.81 (7.27)		55.65 (7.15)	55.92 (7.67)	59.6 (9.07)	0.001
Psychological	55.92 (13.26)		58.29 (10.59)	59.93 (14.25)	54.93 (8.69)	0.001
Social	62.06 (15.72)		57.34 (15.47)	64.43 (16.90)	56.57 (9.85)	<0.001
Environment	58.34 (11.69)		57.57 (13.46)	60.02 (12.54)	59.55 (9.25)	0.465
Pairwise Comparison						
	1–2	1–3	2–3	1–4	2–4	3–4
Overall QoL	ns	***	**	***	***	***
Health Satisfaction	ns	ns	**	***	*	***
Physical	ns	ns	ns	***	**	**
Psychological	ns	**	ns	ns	ns	ns
Social	**	*	***	**	ns	***

ns denotes non-significant, * denotes significant level of <0.05, ** denotes <0.01, *** denotes <0.001.

**Table 6 healthcare-10-00523-t006:** The differences in WHOQOL-BREF domain scores across sociodemographic groups.

	Overall QoL	Health Satisfaction	Physical	Psychological	Social	Environment
Gender						
Male	3.01 (1.26)	3.45 (0.79)	58.11 (7.60)	56.99 (12.55)	59.92 (15.18)	60.57 (9.58)
Female	4.05 (0.83)	3.50 (1.13)	53.04 (6.33)	57.81 (13.66)	63.99 (16.61)	56.06 (14.59)
*p*-value	<0.001	0.001	<0.001	0.225	<0.001	<0.001
Age						
≤40 years	3.95 (0.87)	3.62 (1.10)	55.02 (6.88)	58.19 (15.46)	63.58 (17.24)	57.39 (13.94)
>40 years	2.70 (1.27)	3.26 (0.60)	57.52 (8.16)	56.11 (8.32)	58.74 (13.30)	60.64 (8.60)
*p*-value	<0.001	<0.001	<0.001	0.002	<0.001	<0.001
Education						
High school or below	3.34 (1.23)	3.46 (0.92)	56.12 (7.37)	56.92 (12.85)	61.22 (15.51)	58.65 (11.47)
Bachelor or higher	4.30 (0.63)	3.56 (1.18)	55.56 (9.09)	61.38 (13.94)	65.03 (19.12)	59.73 (17.35)
*p*-value	<0.001	0.042	0.382	0.018	0.014	0.668
Marital status						
Married	3.60 (0.68)	3.31 (0.98)	52.52 (6.60)	56.85 (13.58)	58.19 (19.6)	55.96 (12.57)
Single or divorced	3.37 (1.36)	3.53 (0.92)	57.36 (7.74)	57.49 (12.79)	62.79 (14.12)	59.77 (11.78)
*p*-value	0.497	0.051	<0.001	0.056	0.01	<0.001
Occupation						
Employed	3.40 (1.23)	3.46 (0.96)	55.95 (7.39)	57.01 (13.08)	61.38 (16.27)	58.72 (12.17)
Unemployed	3.80 (0.97)	3.53 (0.60)	57.57 (9.17)	61.35 (11.28)	63.93 (9.47)	59.09 (11.30)
*p*-value	0.067	0.447	0.12	0.15	0.108	0.606
Monthly income						
<5000 SAR	3.39 (1.26)	3.41 (0.93)	54.97 (7.50)	56.43 (12.22)	60.05 (16.56)	57.94 (11.41)
≥5000 SAR	3.50 (1.16)	3.56 (0.95)	57.76 (7.28)	58.69 (14.04)	63.90 (14.53)	59.99 (13.02)
*p*-value	0.06	0.007	<0.001	<0.001	0.002	<0.001
Smoking						
Smoker	3.80 (0.92)	3.53 (0.73)	56.02 (7.36)	58.01 (17.20)	67.01 (14.27)	61.75 (12.74)
Non-smoker	3.31 (1.28)	3.45 (1.0)	56.08 (7.60)	57.09 (11.27)	59.75 (16.0)	57.75 (11.72)
*p*-value	<0.001	0.038	0.484	0.338	<0.001	<0.001

Comparisons were performed using Mann–Whitney U test.

## Data Availability

The datasets generated for this study are available on request to the corresponding author.

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
