# Peer review of "Quality of Life among Caregivers of Patients Diagnosed with Major Chronic Disease during COVID-19 in Saudi Arabia"

_healthcare, 2022, doi:10.3390/healthcare10030523_

Round 1

Reviewer 1 Report

The article talks about an important issue which is the burden that caregivers experience when they take care of patients of chronic illnesses. The strong/positive points of this study is the large number of participants, extensive data analysis ( as shown by the results) and extensive literature review including discussion. However, my enthusiasm about this paper is dampened by some serious flaws.  Specifically here are some weak points of the manuscript:

  1. Abstract: In line 29 the authors talk about behaviors? What behaviors? This is not clear in the abstract session.

  1. Introduction:

            * The authors use often the word “ normally”. I would substitute the word with the words “traditionally” or “usually”

            * The authors make lots of claim without supporting evidence in other words include a reference. For example ,lines 58, 59 after the word “annually” there should be a reference. Also in line 49, after the word “studies” or at the end of the sentence there should be some references supporting the statement.

* There are a lot of typos and syntax errors (e.g. line 82 “vulnerable”, or line 63 the word “additionally” is not needed. Also at the end of the sentence ( line 64) there should be more references.

* Overall, most of the literature cited in the introduction came from studies that took place in the western world. What about studies that took place in the GCC countries? If there is limited literature coming from this region it should be noted in the manuscript.

* What research gap is the manuscript and study trying to address? Why is the study so important?

* What  are the research questions the study is trying to answer?

  1. Materials and Methods

 * Has the tool been pretested among a group of caregivers in Saudi Arabia? Has it been translated into Arabic or was it administered in English?

* What are the psychometric properties of the instrument?

* Who are the caregivers? This is a big flaw in the study? Are the caregivers the family members of the patients? Through my experience living in a GCC country, most families have a helper ( from another country) who helps with cooking, transportation and even raising the children. It is not uncommon that this person also helps with taking care of a member of a family who suffers from a chronic disease. This is not clear in the manuscript. Again, who are the caregivers who answered the survey? In addition, there is no information how much time they spend on a weekly or daily basis taking care of the patient. I think the amount of time does make a difference in their overall social/emotionally wellbeing. What does the current literature say on this?

* It looks like the nurses acted as researchers and collected the data online. Were the nurses trained by the research team how to collect the data?

* Was the study approved by an ethics committee from an academic institution in S. Arabia? The authors do mention that they have obtained an IRB approval ( line 152) but they do not mention from where. There is a difference between an implied “consent” (line 157 )and an INFORMED consent. There must be an ethics protocol in place and the nurses ( who are doing the data collection) should have been trained in ethics and how to conduct research in an ethical manner.

* What was the response rate? How were the missing values treated? I suggest that the authors use the STROBE guidelines to write the methods section.

* There is no information on data analysis.

* Were the hospitals involved in the study private or governmental hospitals?

* Was the sample a convenient sample? Why not use a random sample? Where there any selection criteria of the participants?

  1. Results

* The results look fine but if there is no hypotheses or research questions developed as part of the planning of the study it looks like that the authors did some interesting statistical analyses for the sake of the analyses. The analyses should be guided by the research questions or a theoretical framework which is also missing.

  1. Discussion

* I think the discussion is too long. On the other hand, I was glad to see that the authors brought up the cultural perspective in regard to the family structure in S. Arabia.

* Line 235: the references are outdated. Need to include more recent references.

* So the conclusion of the paper is that overall the quality of life of the caregivers is good. Why? I am not sure as to how the authors justified the reason behind this conclusion. Is it because of the dense social networks in the local society? It is not clear to me.

* The authors state in lines 275 to 281 that there were no differences in gender distribution that this could be attributed to the fact that males with the change in cultural norms in S. Arabic are more involved in the family’s wellbeing. Are there any sociological studies to confirm such claim or is it a speculation?

Reviewer 2 Report

Aljuaid et al. evaluated the QoL of caregivers who guardian patients diagnosed with heart disease, diabetes, cancer, and/or other disease during COVID-19 pandemic. P. Authors revealed that caregivers of patients with cancer reported the highest mean level of QoL, followed by diabetes, cardiovascular diseases, then other different diseases. It is interesting study in hot topic which is COVID-19 pandemic. There were some similar studies, but not performed in so original population. There are some spelling errors. I also suggest to discuss in introduction section potential impact of COVID-19 on neurology caregivers (Neurol Neurochir Pol 2020;54(2):207-208. and Neurol Neurochir Pol 2021;55(4):413-414.). Therefore, I recommend major revision.

Reviewer 3 Report

The manuscript provides interesting insights into other cultural contexts that are not usually addressed in the traditional literature on quality of life. After an in-depth reading of the content, I comment below on some areas for improvement:

  • Lines 24, 29, 136, 149, 159 and Table 2: Please add a comma in the thousands.
  • Line 27: Add a space between "=3".
  • Line 57: Please change "dibabetes millitues" to "diabetes mellitus".
  • Line 82: Please change "vonerable" to "vulnerable".
  • Line 85: (1) Who are Anneke and others? There is no explicit reference in References section. A solution would be to add this reference or to replace the expression "Anneke and others" by "Several authors". (2) In addition, consider using the term "et al." instead of "others" in the manuscript, as this is more common.
  • Lines 94 and 127-128: The font or font size is different from the rest of the text.
  • Lines 95 and 98: "Some studies" is mentioned, but there is only one reference for this statement.
  • Line 109: Please replace "practcally" with "practically".
  • Lines 111-115: This statement is repeated several times throughout the introduction. Consider deleting it or keeping the new information it provides.
  • Line 116: Please change "caregiver burden" to "this condition", as in the previous line the same expression is repeated.
  • Line 117: There are no previous references to these authors. I suggest this modification: "Another paper about the prevalence..." and include their reference in the manuscript.
  • Line 124: I suggest changing "who guardian" to "of".
  • Lines 128-130: There is no need to add another objective when the main objective of the study was stated a few lines before. I propose to remove the objective from these marked lines and integrate the new information into the objective of the study.
  • Line 130: (1) Please change "The" to "This". (2) Although it is mentioned in line 159, it should be noted here that it is a cross-sectional study. (3) Have you used the STROBE checklist for cross-sectional studies (https://www.strobe-statement.org/checklists/)? Although the journal does not consider it a prerequisite for the submission of this type of study, please consider reviewing especially items 9 and 10 of this checklist.
  • Line 133: There is a typo in "questoins".
  • Line 134: Other typos ("questionnair" and "throuhg").
  • Line 148: Which "other diseases" have been included? These are supposed to be other chronic diseases, but their chronic character is not mentioned. Also, it is important to mention at least the most representative ones within this group, as they are then compared with the other 3 groups. Without specifying, it would be a bias of the study itself, as it is not known which diseases are being compared (especially when many significant differences are watched when comparing this group with the others in Table 5).
  • Line 149: Were the 2,000 caregivers included in the study the result of a sample size calculation? If yes, please indicate the parameters set in that calculation.
  • Line 152: There is a typo in "Instituional".
  • Line 156: Another typo ("acceptence").
  • Line 157: Please do not explain in Results section, both in the text (lines 180-181 and 209-212) and in the table captions (1, 2, 3 and 6) the procedures of the statistical analysis. You should include a separate subsection (Statistical analysis) in the Material and Methods section.
  • Line 159: Another typo in "registerd".
  • Lines 161-162: Please consider changing "40.5% were less than 30 years" to "less than 30 years (40.5%)" to maintain the order in which the data are written.
  • Line 163 and Table 1: SAR or SR? Use the same abbreviation.
  • Tables 1 and 6, and lines 276, 312, 321, 329 and 403: As described in the "Instructions for Authors" section of Healthcare (Sex and Gender in Research), you should change "gender" to "sex", as the entire study mentions the biological dimension of the variable.
  • Lines 170-171: Please add the word "points" after standard deviations.
  • Table 2: I believe this table could be improved to better show which data are frequencies, percentages and means.
  • Line 197: Please remove the comma.
  • Lines 240-242: I suggest the following sentence: "...experience: (1) the gratitude for being able to help those in need, (2) their sense of accomplishment, and (3) realization of the significance of good health [18]."
  • Line 256: There is a typo in "Ccardiovascular".
  • Line 264: Another typo in "evethough" (two words).
  • Line 266 and 397: Please reconsider "(2006)" and "(2014)", as this expression has not been used previously in the manuscript.
  • Line 288: Remove one "health".
  • Lines 297-298: This sentence is incomplete or needs to be grammatically checked. I suggest deleting "Although".
  • Lines 314-317: After reading the results, I think the main cause is the cultural difference of the participants between the previous studies and the one in this manuscript (as mentioned later on in lines 377 and 428).
  • Line 387: Please replace the full stop with a comma.
  • Line 388: I suggest changing "A better financial situation" to "This financial situation".
  • Line 389: Please change "the caregiver" to "caregivers".
  • Lines 391-392: Please correct the typos "sennce" and "burdenedsome".
  • Line 395: Please remove "the".
  • Lines 410-411: If the most frequent chronic diseases cannot be mentioned under "other diseases", it should be added as a limitation.
  • Line 436: Please remove "Please add:".

The authors propose a cross-sectional study on quality of life in caregivers of chronically ill patients in a Muslim cultural context (Saudi Arabia), In fact, there are differences in the results with respect to other similar studies conducted in other countries (mainly in America and Europe).

Although there are several errors in the wording of the text and some structural issues such as the need for a statistical analysis section in the Material and Methods section, the analysis and content itself are quite correct, so it is a manuscript that can be substantially improved after the corrections to be made by the authors. I am particularly concerned that (1) there is no mention of a sample size calculation and (2) chronic diseases under "other diseases" are not specifically defined.

I hope that my remarks will help you to improve the manuscript.

Best regards.

Round 2

Reviewer 2 Report

Authors improved article according to my comments. I have no further remarks. I suggest acceptance of the paper.

Reviewer 3 Report

Thank you for the thorough review. There is an improvement in the content and wording of the text. Due to the PDF export form of the manuscript, I cannot see all the comments, but I read the responses included in the attached file.

Furthermore, I add some observations that could improve the text of the manuscript:

  • Line 79. If you include the unit (0.) in p-value and r-value, consider including it in all p-values and r-values in the rest of the manuscript (i.e. line 563).
  • Lines 182-256. The reference numbers are repeated. The same occurs in other lines of the manuscript. Please check this detail.
  • Line 261. "disease advance" seems to have a different letter font.
  • Lines 287-289. The main objective should appear at the end of the Introduction. It is strange to read it in the middle of this section.
  • Line 301. I think this line is in bold.
  • Lines 362 and 378. Please, consider changing "investigation" to "research". It is more specific in this context.
  • Lines 379-380. If you follow the STROBE checklist, this information is not necessary.
  • Lines 395 and 403. Please, consider changing "investigators" to "researchers". It is more specific in this context.
  • Line 405. Please, consider changing "Data" to "data".
  • Lines 479-480. The data on participants has already been discussed above, on lines 407-408. I suggest that this data only be in Results.
  • Tables 1-6. Remove legends below the tables, at least the descriptive statistics, as it is indicated in the column labels.
  • Lines 1,013-1,020. Please, add the reference to this study.
